# Rights–Values–Interests: The Conflict between World Cultural Heritage and Community: A Case Study of the West Lake Cultural Landscape Heritage in China

**Jiancheng Lu [1], Xiaolong Luo [1,\*] and Peigang Zhang [1,2]**

[1]  School of Architecture and Urban Planning, Nanjing University, Hankou Road No.22, Nanjing 210093, China
[2]  Jiangsu Institute of Urban Planning and Design, Caochangmen Road No.88, Nanjing 210036, China
\*  Correspondence: lujcnju@126.com

**Abstract:** The conflict between world cultural heritage and local communities is investigated by using the cultural landscape heritage of West Lake in China as a case study, and establishing an analytical framework of "Rights–Values–Interests" based on the property rights theory of the new institutional economics and the value and interest structure characteristics of cultural heritage. The conflict problem in the market environment is analyzed based on a theoretical explanation. An in-depth discussion of the framework and improvement of China's protection institution is provided. We outline the following key points: First, the Chinese government "plundered" certain behavior rights and legitimate interests of community residents through the enactment of protection laws, leading to a conflict between the protection and community. Second, China's laws lack a clear definition of the power and responsibility of the central and local governments with regard to protection actions, leading to vague positions of the government and exacerbating conflicts. Third, China's protection laws are out of touch with the laws of private property rights. The root cause of the conflict is that the protection action only considers the protection law as the core but neglects the residents' legal behavior rights. Finally, from the perspective of considering the residents' legitimate interest demands, defining behavior rights boundaries, and strengthening administrative management, we propose to improve the protection institution in order to achieve the harmonious integration of heritage protection and local communities, and we call for a greater focus on the legitimate interests or survival rights of ordinary Chinese community residents.

**Keywords:** rights–values–interests; world cultural heritage; local community; conflict; West Lake cultural landscape heritage

## 1. Introduction

"West Lake (Hangzhou, China) belongs to the Hangzhou people but also to Chinese and foreign tourists. The successful designation of the West Lake as a World Heritage Site has effectively protected the natural landscape and historical cultural relics, enabling Chinese and foreign tourists to better get close to and experience the West Lake Area." (Secretary of the Hangzhou Municipal Party Committee, February 2009)

"After becoming a world heritage site, the influx of tourists to the West Lake area is also fueling the city's economic growth. In 2018, West Lake received 28.14 million tourists annually, ranking first among all cultural heritage sites in China." (Excerpts from the Hangzhou Daily, March 2019)

"We are the current residents of West Lake. Our communities are located in the West Lake Heritage Area. Most of the community buildings were built in the 1970s and 1990s.

They are badly damaged and the support service facilities are not complete. In addition, the government's heritage protection laws prohibit repairing the building and there are frequent leaks and power outages." (Excerpts from interviews with community residents, May 2018)

These three statements indicate that different groups of people have large differences in their views regarding the protection of the West Lake Cultural Landscape Heritage (WLCLH) area in Hangzhou, the capital of Zhejiang province in China. Since the establishment of the West Lake Scenic Area (WLSA) in 1982 to the successful designation of WLCLH in 2011, the Hangzhou municipal government has always attached great importance to the protection of the area and was determined to make it a tourist destination for both Chinese and foreign tourists. According to media reports, the Hangzhou municipal government has achieved its stated goal of increasing the area's popularity and the number of visitors. At the same time, the local community residents of WLCLH have different views on the protection of the site and believe that the approach to protection has a negative impact on their basic life needs.

In the 1980s, the emergence of heritage conflicts involving human rights and social equity resulted in a large number of scholars becoming increasingly interested in community studies [1]. English Heritage (1997) mentioned that the daily living environment of community residents may be the basis for the distinctiveness and identity of cultural heritage [2]. Community participation is an important prerequisite for "representative heritage" [3]. The concept of "community residents have the right to their own historical and cultural heritage" elevates the status of community participation [4]. At the same time, some scholars suggested to create a dialogue platform between cultural heritage protection and community development and to negotiate the disharmonious views and conflicts between them [5,6].

However, before the 1990s, such open conflicts between protection and development were not common under China's pro-growth authoritarian regime [7], because the central goal of the Chinese government's policymaking has always been economic growth and the attempt to escape poverty and backwardness [8]. Other opinions related to heritage preservation that may affect the hegemony of "development" are rarely expressed [9,10].

Since the beginning of this century, conflicts between cultural heritage and local communities have appeared in China and have become increasingly serious. Most of the literature indicates that the appeals of residents in these communities are centered on the redistribution of benefits [11]. Community residents are often defined as pure profit seekers. They want a lucrative compensation scheme [11,12] rather than blocking the government from protecting it. This kind of struggle aimed at pursuing interests hardly affects the unequal power structure of fait accompli, nor can it fundamentally change the political marginalization of community residents [11]. The existing literature suggests that local community residents play games with the government in order to gain more benefits, but mostly ignores the benefits that the community residents are fighting for. Additional benefits or legitimate benefits that should be theirs are being taken away by the power of the government. In the Chinese context, what are the root causes of the deprivation of the interests of local communities?

In order to clarify and answer the above question, the background of protection and cultural heritage characteristics of China must be integrated. Obviously, market economy thinking has penetrated into all aspects of social life. When social contradictions caused by the protection actions keep emerging, it is necessary to explore the causes of the contradictions from the perspective of the interests and rights of the "people" behind the material structure of cultural heritage, and find solutions to the problems. First, what is the motivation of the Chinese government (officials) to protect world cultural heritage (WCH)? Second, what is the relationship between the value composition of WCH and the interest of community residents? Third, in the protection of WCH in China, how does the protection law deprive ownership and behavior rights of community residents?

Based on the theory of property rights, this study establishes an analytical framework of "right–values–interests" and uses the WLCLH in China as an example to analyze the value composition and interest structure behind the material structure of the WCH and to explore the causes of conflicts

from the perspective of economics. In addition, this study discusses the improvement of China's protection institution.

## 2. Literature Review and Theoretical Framework

### 2.1. Literature Review: WCH and Local Communities

The role of community in heritage has always been a hotspot in the field of international heritage protection [13–16]. The Recommendation Concerning the Safeguarding of the Beauty and Character of Landscapes and Sites, which was adopted by UNESCO in 1962, first mentioned the concept of community in heritage documents [17]. Since then, community has been a common keyword in documents such as the Budapest Declaration, World Heritage Convention, and Nara+20 [18].

In 1992, UNESCO changed the focus of world heritage from heritage itself to the role of people in the World Heritage Convention [19]. In fact, according to some advocates, organizations such as ICCROM have come to recognize the need to integrate community members when preserving historical cultural heritage [20]. At the same time, appropriate campaigns should be conducted to improve the quality of life in the community [21]. In addition, the Valletta Principles further emphasize the protection of local communities and the preservation of their traditional and unique living habits [22]. Communities are encouraged to participate in heritage management projects in an "ecologically and culturally sustainable" manner [23]. It can be seen that heritage protection has been incorporated into the framework of sustainable development, which not only emphasizes heritage itself, but also emphasizes the improvement of social cohesion and quality of life of community residents [24]. This situation reflects the humanistic development of WCH from a material structure-based approach to using social values as the core [17]. The improvement of community status promotes the mutual benefit of cultural heritage protection and the interest of community residents [25].

Although according to UNESCO documents, communities are officially included in the protection of cultural heritage and are respected [18], conflicts between heritage protection and community development are still common around the world [26]. Conflicts often focus on whether to involve the community in heritage practice [27]. Some experts and scholars have called on the community to participate in the planning and management of heritage, including management decisions and the distribution of benefits [14,16,27,28]. However, the interests of local communities are often overlooked by heritage managers, despite their important role in the heritage [29]. In addition, there are often conflicts between heritage preservation requirements and community development [30]. In order to achieve heritage protection, economic reciprocity, and socially acceptable win–win cooperation, the needs of local communities should be taken into account in protection law [31]. Although it is difficult to meet everyone's needs, greater consensus should be achieved [32]. However, from the perspective of reality, cultural heritage and community conflicts have been effectively alleviated in developed countries in Europe and in America [33], but are still common in developing countries [34].

In China, where administrative power is highly concentrated, the conflict between heritage protection and community development is faced with numerous challenges [35,36]. Since the founding of the People's Republic of China, the power of economic development has been gradually delegated to local governments, but central political power has been consolidated [37]. The word "community" has long existed in China but it has not been incorporated into the national governance system or involved in the decision-making process [38]. At the national level, there is little incentive for communities to participate in heritage conservation [39]. At the local level, the rise and fall of central government leadership depend on whether economic growth is achieved [37], and local government leaders often focus on promoting urban economic development [39]. As a result, China's heritage protection laws often ignore community interests [13]. In this context, community residents may suffer from poor government consideration and lack of a compensation system [40,41]. The way of life, employment, and the social security system may change, causing social dislocation, income reduction, and other problems [42,43]. Therefore, heritage protection needs to consider the development demands of local

community residents, such as politics, economy, and culture [40,44]. Residents should be compensated when their interests are not considered [43].

Although many experts have put forward suggestions with regard to strengthening community participation and improving compensation mechanisms, can the conflicts and contradictions behind the protection of cultural heritage be solved by formulating and improving the laws on the protection of cultural heritage? If so, why is there still a contradiction between "legal rights" and "practical interests" in China's current protection practice? It can be seen that it is difficult to solve the contradictory problems associated with protection simply by improving protection law. It can be argued that the perspective of the problem analysis should be shifted from the material structure of WCH to the "people" behind it and to the actual economic relations in order to reconsider the improvement of the protection system.

*2.2. Theoretical Framework*

2.2.1. Value Composition and Interest Structure of WCH

WCH is composed of multiple values, including culture, history, economy, ecology, and society [45]. Some of these values are inherent in the heritage itself, while others are authorized by certain agendas, as in the Authorised Heritage Discourse coined by Laurajane Smith [46]. These authorized values constitute the legitimate and legal protection of WCH. From a social perspective, the scarce and irreplaceable historical, cultural, and social values are the core values of WCH, which can be called heritage value. The formation of heritage value includes three stages: First, social goods are produced by necessity. Second, it has significance and value beyond its original purpose. Third, it is determined to be a WCH, forming a heritage value authorized by professional elites (state officials in China) [47]. It can be said that the value of heritage is the special value of cultural heritage resources endowed by the social collective representative (elites and officials). This process also confirms the public attribute of heritage value; that is, the social collective jointly owns the heritage value. In other words, WCH is protected because of public interest.

However, WCH also contains contradictory values [48]. The aforementioned discussion indicates that WCH is originally a social product produced by human needs that consumes and occupies certain resources and thus has economic value. For example, as a part of the material space of WCH, the community requires a certain amount of land, material, capital, and labor resources for its construction and can achieve economic use by living, trading, and renting. At the same time, it also constitutes an important part of individual ownership. Although people do not build communities only for economic purposes, they constantly transform external resources into internal economic values during the process of construction, use, and maintenance, which constitutes the original value of WCH. Therefore, WCH must first have an original value. However, the Chinese government continues to emphasize the significance of the preservation of heritage value through top-down management, but exerts less effort to maintain the original value of community residents. As Rypkema points out, in the long run, the economic impact of WCH is far less than its impact on education, environment, culture, and society [45]. But the economist John Maynard Keynes said, "in the long run, we are all going to die". In the short-term, in China, community residents are more concerned about whether WCH has a negative impact on their basic living needs and economic income. Therefore, to some extent, WCH also represents the private interests of local community residents [49].

2.2.2. Property Rights Theory of the New Institutional Economics

In the legal context, property rights in the west have a similar meaning as Chinese ownership. The property rights in new institutional economics are essentially different from this definition. Property rights theory holds that the property right of a resource is a collection of behavior rights, and what is traded in the market is not the resource itself but the transfer of rights [50]. Its concern is not the relationship between people and resources, but how people acquire and exercise their behavior rights to

the resources [51]. It determines the behavior norms of individuals in the face of resources. Individuals must abide by the relationship between people or bear the cost of not honoring the relationship [52]. When the conflict of interest in the use of resources is caused by scarcity, it must be solved through the code of conduct, namely, the institutional arrangement [53]. The definition of rights is the process of institution making; that is, through a legal mechanism or negotiation mechanism, the rights of behavior of resource utilization are clearly allocated to each subject and the boundaries of the rights of behavior are defined [49].

WCH is a resource with commodity characteristics [54]. It contains a set of behavior rights and the process of protection involves the transfer of rights [49]. If the national public power does not intervene in the protection action, the protection subject wants to obtain the protection behavior right only through the market transaction to gain its ownership. For example, in the protection of Mount Vernon, the Mount Vernon women's association founded by Ann Pamela Cunningham first acquired the ownership from the Washington family and then had the right to protect the area [55]. With the development of society, the government's public intervention is accepted by society and the government can set the behavior rights and responsibilities of protection by promulgating protection laws. Therefore, it can be argued that the conflict in protection is not unclear ownership, but the unclear definition of the right to use, the right to development, and other behavior rights [56]. WCH needs to clearly define the boundary of stakeholders' behavior rights and separate the public and private parts of the heritage through institutional construction [57].

There is a consensus that property rights theory is important in the field of cultural heritage protection [58–60]. An analysis based on behavior rights is very necessary for cultural heritage protection. However, protection is not a simple economic behavior, and cultural heritage is not a common public resource. It can be argued that the discussion from the perspective of property rights theory should be based on the institutional characteristics of cultural heritage and laws that specifically embody property rights. Otherwise, it becomes a theoretical explanation of relevant issues and it is difficult to provide guidance for practical applications.

Therefore, it can be considered that the behavior rights of the property rights theory as the core combines the value and interest characteristics of cultural heritage to establish an analytical framework of "rights–values–interests" (Figure 1). At the same time, the discussion of legal ownership is extended. The WLCLH is used as an example to determine the evolution of the relationship between property ownership and behavior rights, values, and interests in local communities in the process of determining the protection behavior rights of the Chinese government.

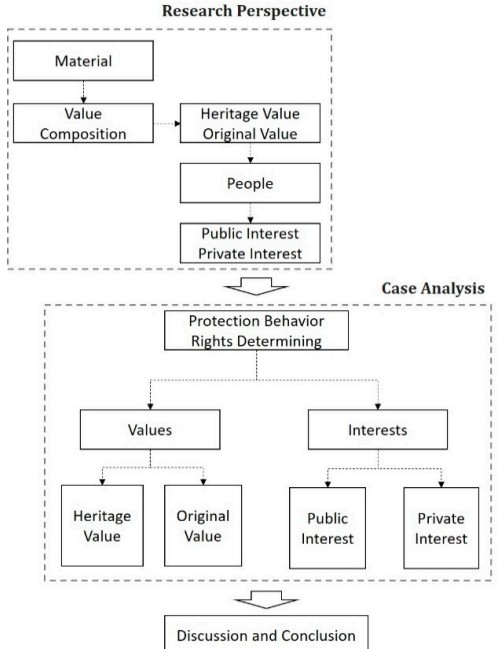

**Figure 1.** "Rights–values–interests" analysis framework.

## 3. Study Area and Methods

### 3.1. Study Area: WLCLH in Hangzhou, China

West Lake is located in the west of Hangzhou City in Zhejiang province in China. In 1982, it was rated as one of the first national scenic areas and was officially included in the Chinese heritage protection system. In June 2011, it was officially listed as a World Heritage Site. At present, the WLCLH is managed by the Hangzhou WLSA Management Committee. There are nine communities in the WLCLH (Figure 2), namely, Maojiabu, Shuangfeng, Longjing, Wengjiashan, Manjuelong, Yangmeiling, Meijiawu, Jiuxi, and Fancun. As of May 2018, the nine communities had a total population of about 6900 people. These nine communities were selected as research objects because they all have a history of more than 100 years and they have undergone a change from development area communities to protection area communities. Therefore, the investigation and analysis of the dynamic problems faced by the nine communities can provide insights into the impact of WCH protection on local communities.

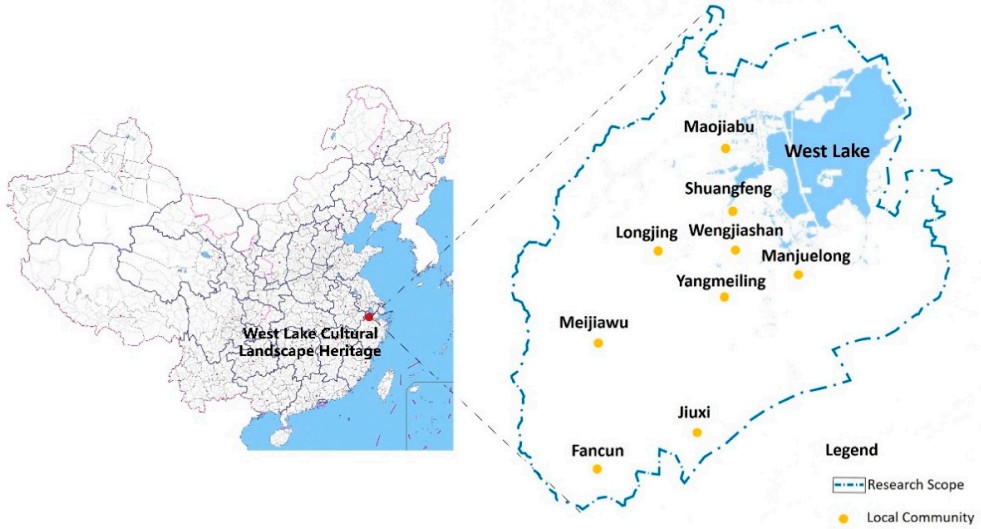

**Figure 2.** Location of the West Lake Cultural Landscape Heritage (WLCLH) and communities.

*3.2. Research Methods*

This study integrates qualitative and quantitative methods for data collection and analysis, which have been widely used in the study of heritage protection and local community issues [61]. A questionnaire survey and key person interviews are the main data collection approaches. Field observations and government statistics are secondary approaches.

The first author once participated in the application for WLCLH during the summer social practice in the university (working time: April to August 2010). After graduation from graduate school in 2014, the author worked in the Hangzhou Cultural Relics Protection Department and participated in the compilation of the WLCLH Protection Plan and Community Development Plans (working time: 2014–2018). During the four years, the author collected a large amount of research material.

In addition, in order to obtain additional information for this study, the author conducted three field surveys in May, June, and July 2018. The first survey lasted 5 days and consisted of semi-structured interviews with the Zhejiang Provincial Bureau of Cultural Heritage, the Hangzhou Xihu District Government, the WLSA Management Committee, etc. and relevant data files and statistical data were collected. Interviews with government departments at all levels included the laws and regulations, protection motivation, administrative measures, and community development issues in the management of the WLSA and the WLCLH.

The second survey lasted for 9 days and consisted of semi-structured interviews with managers and residents of each community, among which no less than five managers and no less than 12 residents were required for each community (Table 1). Since the interview questions were related to the community development conditions from the 1970s to the 1990s, it was stipulated that the proportion of residents over 60 years old interviewed in each community should be at least 30%. In order to increase the authenticity and the comfort of the people during the interviews, the interviews between community managers and residents were conducted at different times and places with an average time of 15 min per person.

**Table 1.** Number of community managers and residents interviewed in different communities (*N* = 169).

| Communities | Number of Community Managers | Number of Community Residents | | | | |
|---|---|---|---|---|---|---|
| | | 18–30 Years Old | 30–40 Years Old | 40–50 Years Old | 50–60 Years Old | >60 Years Old |
| Maojiabu | 7 | 1 | 2 | 2 | 3 | 4 |
| Shuangfeng | 5 | 1 | 2 | 2 | 3 | 5 |
| Longjing | 5 | 2 | 2 | 3 | 2 | 5 |
| Wengjiashan | 6 | 3 | 1 | 3 | 3 | 4 |
| Manjuelong | 7 | 0 | 2 | 3 | 2 | 5 |
| Yangmeiling | 5 | 3 | 2 | 3 | 2 | 4 |
| Meijiawu | 6 | 1 | 1 | 2 | 3 | 5 |
| Jiuxi | 5 | 2 | 1 | 3 | 3 | 7 |
| Fancun | 5 | 1 | 1 | 2 | 2 | 5 |

The third survey lasted 5 days and consisted of on-site questionnaire surveys of community residents (Table 2). The questionnaire design was divided into four parts. The first part consisted of the basic personal information of respondents, such as age, income, and gender. In the second part, interviewees were asked to talk about the impact of the scenic area and WCH on rights and interests of residents, including positive and negative impacts, especially the impact of WLCLH on the residents' ownership and behavior rights. The third part was an evaluation of the community managers' management behaviors to obtain their daily work contents. The fourth part was an evaluation of the WLCLH manager, such as protection motivation, conflict management, and interest compensation.

**Table 2.** Number and proportion of questionnaires in each community (*N* = 740).

| Communities | Questionnaire Number | Proportion | Communities | Questionnaire Number | Proportion |
|---|---|---|---|---|---|
| Maojiabu | 68 | 14.23% | Yangmeiling | 63 | 16.94% |
| Shuangfeng | 89 | 10.85% | Meijiawu | 103 | 12.12% |
| Longjing | 127 | 11.19% | Jiuxi | 41 | 30.83% |
| Wengjiashan | 109 | 10.23% | Fancun | 59 | 14.05% |
| Manjuelong | 81 | 10.71% | | | |

In order to improve the questionnaire participation rate, working days in May, June, and July 2018 (the least busy days in the off-peak season) were selected as the survey time. Finally, 740 valid questionnaires were collected in the nine communities. The response rate was not high but each community's response rate was more than 10% and there were no significant response biases. This study only deals with data related to community issues resulting from the establishment of the WLCLH.

## 4. The Dynamic Process of Conflict between WLCLH and Local Communities

### 4.1. The WLSA and the Local Communities Were Harmonious and Mutually Reinforcing

In 1982, the WLSA in Hangzhou was officially incorporated into China's heritage protection system. Influenced by the political movement of the Cultural Revolution (1966–1976), the environment and cultural relics of West Lake and the surrounding areas suffered significant damage. According to three community residents over the age of 60, it was difficult to encounter green vegetation and clear water at that time, which affected the life of the community residents.

In order to improve the negative effects of a previous political campaign, the Hangzhou government proposed the WLSA renovation project, which is listed as a nationally important project. The Hangzhou Cultural Heritage Bureau was responsible for the project. From 1982 to 1984, projects such as lakeshore construction, sewage treatment, forest restoration, and lakeshore park construction were performed successively. After about two years, the overall environment of the WLSA had taken on a new look. According to the project management personnel at that time:

"Two reasons for the efficient completion of the WLSA renovation project: one is to respond to the call of the state to establish a scenic area and complete the national assessment. Second, the Hangzhou municipal government has performed a good job during the renovation of the WLSA and has made it a famous tourist attraction, which continuously improves Hangzhou's popularity in the country and even the world. Meanwhile, the influx of a large number of tourists also brings great impetus to the economic development of the city." (Interview, June 2018)

Although the government's top-down project to improve West Lake focused on the improvement of the public environment, it did not involve the improvement of local communities. However, the gradually-improved ecological and cultural environment provided a good living atmosphere for the residents of the community. At the same time, it also attracted a small number of tourists to visit the WLSA, providing some business opportunities for community residents. According to the residents of the Maojiabu and Shuangfeng communities:

"The communities around the lake have benefited financially by converting their houses into tourist shops." (Interview, June 2018)

According to the tourism leader in charge of the Hangzhou Cultural Heritage Bureau, the annual number of tourists in 1984 was about 260,000, which was 180,000 more people than in 1982. The increase in tourist numbers has also led to the city's economic growth, making the Hangzhou city leaders more determined to develop tourism around the WLSA. Therefore, the Hangzhou City Master Plan

compiled in 1984 mentioned that "the protection and tourism development of WLSA" is the primary goal of Hangzhou and defines Hangzhou as a "scenic tourism city". The plan put forward five goals for the WLSA: (1) Improve the water quality and expand the park around the lake; (2) beautify the mountain forest; (3) repair the temples; (4) repair ancient tombs of famous people in all dynasties; and (5) build sightseeing lanes and walking trails. In the later stage, due to the poor appearance and environment of the community buildings, which was pointed out by tourists, the viewing experience of the tourists was significantly affected and the community buildings and environmental improvement were also included in the planning. However, according to the interview between community managers and residents:

> "Although community improvement has been included in the planning and implementation project, it has not been carried out at a large scale and only the buildings in the community around the lake that affect leisure and enjoyment have been improved. However, other communities have planted tall trees to cover the buildings without fundamentally improving the quality and environment." (Interview, June 2018)

Although the government has not taken direct measures to support the community, the development of tourism has stimulated the entrepreneurial spirit of local residents, and a large number of residents have expanded their houses and transformed them into shops providing services for tourists. According to the questionnaire survey and interview, the building density of the nine communities increased 0.44 times from 1982 to 2001 and 76% of the expanded building space was used for catering, teahouses, shops, and homestay operations. In terms of annual income, in 1982, the average annual income of local residents was between 1000 ¥ and 3000 ¥ ($146–437), whereas, in 2001, the average annual income of residents was between 20,000 ¥ and 60,000 ¥ ($2913–8740); the growth rate was 2.7 times that of the average level of Hangzhou residents.

To sum up, at this stage, the Hangzhou municipal government conducted protection and tourism development of the WLSA, achieving overall environmental improvement and sustainable and stable development of tourism. Although there are no direct support measures for the communities, tourism has increased the income of local communities. It can be seen that the Hangzhou government has used behavior rights to protect the WLSA but this has not affected the ownership and behavior rights of local community residents, nor has it damaged the original value and private interests. It has achieved the harmonious coexistence of heritage value and original value, and the mutual promotion of public and private interests.

### 4.2. Emergence of Conflict Between WLSA and Local Communities

In 2001, the Hangzhou municipal government proposed to declare the WLSA as a World Heritage Site and set up the WLSA management committee, which was responsible for the declaration of WLCLH; members of the management personnel were also members of the Hangzhou Cultural Heritage Bureau. The application work was divided into three parts: (1) To manage the WLSA comprehensive protection project; (2) the compilation of the "WLSA Master Plan"' and (3) to advance legislation and promulgate the "Protection Regulations of the WLSA".

The WLSA comprehensive protection project is similar to the WLSA renovation project mentioned above, which lasted four years and cost 5 billion (RMB). The project included the improvement of the core scenic area (2003), the clean-up of the lake (2003), the beautification of 15 scenic spots (2004), and the restoration of 36 cultural landscapes (2005). The interview between the project leader and the community residents indicated that they generally felt that:

> "The comprehensive protection project of WLSA is very effective. From 2002 to 2005, the number of domestic and foreign visitors increased by 18.71%. Nine communities saw annual per capita income increases by 20%." (Interview, June 2018)

However, the WLSA Master Plan and Protection Regulations of the WLSA promulgated in 2005 have issued some requirements on community control (Table 3), which is generally regarded as the

starting point of the conflict between heritage protection and local community development. The WLSA Master Plan calls for the gradual relocation of existing communities. According to the interview, in 2005, 1037 households in the nine communities were forced to move out of the WLSA because the dilapidated community buildings affected the overall tourism environment and the selection of world cultural landscape heritage. However, most of the residents have been forced to move out and they want to remain in the WLSA. Three community managers commented:

> "Most community residents do not agree with the requirement to moving out for two reasons: First, they have lived here for hundreds of years for many generations and everyone has feelings for this place and is unwilling to leave the house left to them by their ancestors. Second, most of the residents rely on agriculture or tourism to make a living and if they had to leave the WLSA and integrate into city life, they would lack the skills to sustain themselves." (Interview, June 2018)

**Table 3.** Community management requirements in the law and planning of the West Lake Scenic Area (WLSA).

| | Specific Content of Management Requirements |
|---|---|
| WLSA Master Plan | 1. The height of community buildings shall not exceed three stories and 12 m. <br> 2. Strictly control the number of communities and encourage gradual moving |
| Protection Regulations of the WLSA | 1. It is forbidden to build, rebuild, expand, or repair community buildings in WLSA. <br> 2. The community approved for reconstruction and the area of the reconstructed building shall not exceed 80% of the original area. |

The Protection Regulations of the WLSA state that "it is forbidden to build, rebuild, expand, or repair community buildings in the scenic area". Most of the community buildings in the WLSA were built in the 1970s and 1980s, while the life span of Chinese folk houses is generally 10–20 years. In addition, a large number of folk houses serve tourists and overuse accelerates the deterioration of houses. According to the statistics of various communities, in 2005, the average dilapidated housing rate in the nine communities was about 6.8% and the implementation of the ban on repairing community buildings led to an increase in the average dilapidated housing rate to 10.72% in the nine communities in 2010 (Figure 3). According to the residents of the community:

> "The ban on repairing community buildings has had a negative impact on the buildings. On the one hand, as a result of building disrepair, decay gradually occurs in the dilapidated houses and affects our safety. At the same time, the aging infrastructure of power supply and water supply has also affected the quality of life. On the other hand, the disrepair of the houses has also affected the quality of tourism services." (Interview, June 2018)

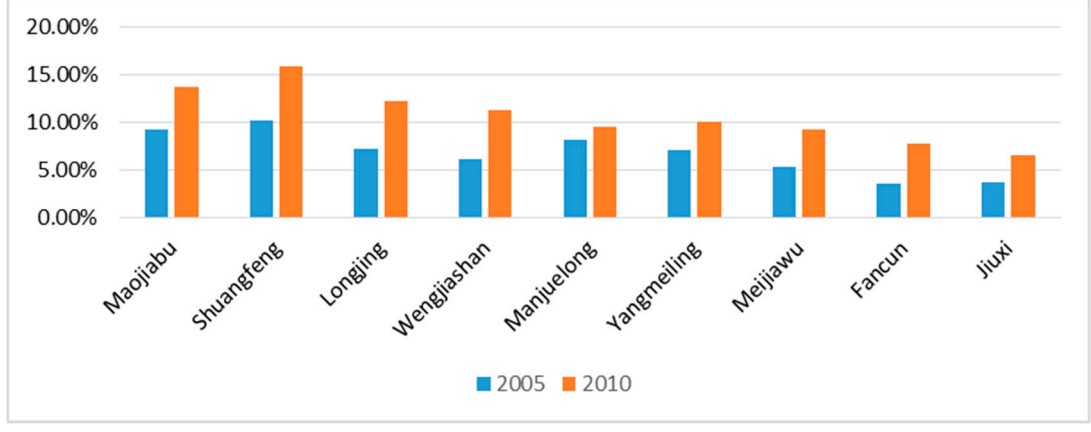

**Figure 3.** Dilapidated housing rate of different communities in 2005 and 2010.

To sum up, at this stage, the WLSA management committee issued the WLSA Master Plan and the protection regulations of the WLSA through top-down hegemony. The committee has outlined the behavior rights of protection of the WLSA. Although this has no impact on the property ownership of local community residents, it restricts some of the behavior rights of residents, such as repairing houses. This action violates the exclusivity of the residents' behavior rights, affects the realization of the original value and private interests, and has negative impacts on the life and employment of community residents.

### 4.3. Intensification of the Conflict between WLCLH and Local Communities

On 24 June 2011, the 35th World Heritage Committee in Paris officially included the WLSA into the World Heritage List. In order to strengthen the management of the WLCLH, the Hangzhou municipal government issued the Hangzhou WLCLH Protection Management Plan and the Hangzhou WLCLH Protection Management Regulations. Some new requirements were added with regard to community control requirements of the WLSA (Table 4).

**Table 4.** Community management requirements in the law and planning of the WLCLH.

|  | **Specific Content of Management Requirements** |
| --- | --- |
| Hangzhou WLCLH Protection Management Plan | Strictly control the amount of construction in the community. |
| Hangzhou WLCLH Protection Management Regulations | 1. Implementation of community building reduction measures. 2. The site selection, layout, height, and volume of community construction projects in the WLCLH shall be coordinated with the surrounding landscape and environment. |

The Hangzhou WLCLH Protection Management Plan stated that "controlling the amount of construction in the community" was required. In view of this control requirement, the WLSA management committee strictly controlled the construction of support service facilities such as schools, medical treatment, water supply, and power supply, which directly affected the basic living quality of local residents. According to the questionnaire of the nine communities, the dissatisfaction rate of residents with kindergartens, primary and secondary schools, and water and power supply facilities exceeded 60%. In the interview, community residents generally mentioned the following:

> "Since 2011, when the construction of support service facilities was controlled, kindergartens and primary schools were gradually moved out. Now children can only go to school outside the WLCLH. The distance is generally 3–8 km, which exceeds the scope of kindergarten and primary school services stipulated by the state and violates the principle of nearby education of the state. Meanwhile, the community water and power supply facilities were built at the end of the last century. The aging facilities make it difficult to meet the needs of residents. In addition to a ban on construction in the WLCLH, about 26% of the community residents now claim that water and electricity are often cut off at peak times." (Interview, June 2018)

The Hangzhou WLCLH Protection Management Regulations put forward "measures to implement the reduction of community buildings". In 2012, the WLSA management committee began to demolish community buildings. They demolished the building that the community had used for tourism development (Figures 4 and 5). By April 2018, a total of 10,500 square meters of buildings had been demolished in the nine communities. As tourism is the main source of income for residents to maintain their livelihood, the demolition of construction space means the loss of income; this is coupled with the difficulty of meeting basic needs of support service facilities and, therefore, the conflict between heritage protection and community development continued to worsen. According to the survey questionnaire, 68.8% of the residents in the nine communities wanted to move out of the WLCLH and 91% of them chose to move out because the heritage protection request infringed on their legal

rights and interests. However, it has been difficult for residents to move out. Government officials are also reluctant to pay additional economic costs while protecting the WLCLH, because relocating local communities does not achieve the goal of economic growth in the city.

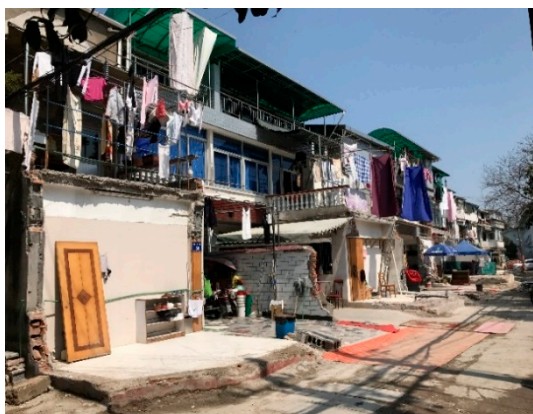

**Figure 4.** Demolition of buildings in Shuangfeng community.

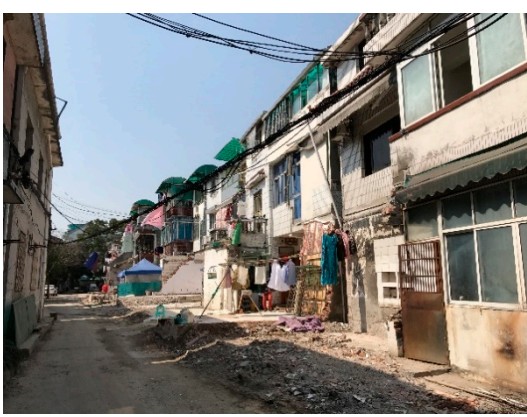

**Figure 5.** Demolition of buildings in Maojiabu community.

To sum up, the WLSA management committee issued the Hangzhou WLCLH Protection Management Plan and the Hangzhou WLCLH Protection Management Regulations. This restricted the construction of community service facilities and buildings that were needed for the residents' livelihood were demolished. In addition to the difficulty in the relocation of the community, the conflict between heritage protection and community is unprecedented. Therefore, the declaration of behavior rights of protection to the WLCLH deprives the residents of their legal behavior rights. This damages the original value of the community and the legitimate interests of residents.

## 5. Discussion

The development process indicates that the conflict between WLCLH and the local community is always dynamic. It can be considered that cultural heritage protection has two types of impacts on the local community. In the first stage, it is indisputable that some protection measures and the development of tourism have also resulted in employment opportunities and income growth for local community residents and a better natural environment. However, as local governments plunder the ownership and behavior rights of local residents by enacting protection laws, their daily basic needs are difficult to meet. As a result, the conflict between WLCLH and the local community continues to deteriorate.

Due to the diverse values and mixed interests of WCH, the protection action involves the interests of different people. WCH and local communities co-exist in the same location, bearing heritage value of WCH and the original value of the local communities, which involves both public and private

interests. Some scholars have indicated that the ownership of cultural heritage is of great significance in the protection process [62]. If the ownership of the overall physical space of WCH belongs to the state, then the original value and heritage value belong to the state and the interests are clear. A good example is the Forbidden City in Beijing. However, if there are private resources in the scope of WCH, such as community ownership in the WLCLH, which belongs to residents, then the original value and heritage value belong to two subjects and their interest structure is relatively complex.

From the perspective of property rights theory, West Lake, whose ownership is not completely state-owned, is recognized as a World Cultural Landscape Heritage site, which means that the community resources whose ownership belongs to the residents also have the behavior right of protection. In this way, the exclusivity of the residents' original behavior right is violated, their legal behavior right is restricted, and the boundary of their behavior is changed, making the community become "common property" shared by residents and the country (social group). This requires a fair and legal redefinition of the boundaries of their behavior rights and thus a widely accepted contract (including community residents).

From experience of developed countries in Europe and America, it can be seen that when people realize that WCH is related to national interests (public interests), the intervention of state power violates the exclusivity of private rights and interests and there exist the basis of public opinion and legitimate reasons. Therefore, through the legal determination of public interest, WCH has the same right to be protected by public power as other public and private interests. If the behavior rights boundary between state and community can be clearly defined and there is a fair mechanism to balance everyone's interest, i.e., the structural balance between public and private interests of WCH can be achieved, then the conflicts of interest in the protection process can be solved. However, as demonstrated in the case study of the protection of the WLCLH, it still focuses on the protection of the material structure of WCH and pays less attention to the interests and rights of the people. Through mandatory promulgation of the legal acts to determine the behavior right of protection, the top-down hegemonic "plunder" of the legal behavior rights of local community residents is occurring. This continues to damage the community's original value and private interests and, coupled with a lack of effective behavior rights definition and interest balance mechanism (compensation mechanism), results in constant conflict between heritage protection and the communities.

## 6. Conclusions

In this study, the WLCLH was used as an example and an analytical framework of "rights–values–interests" based on the property rights theory of the new institutional economics was established to investigate the problems encountered by local communities during the protection process. Different from the Burra Charter [63] and other already-existing models discussed in the heritage literature, the Chinese principles "provide greater room for state intervention in the decision-making procedure". Therefore, this analytical framework emphasizes that state heritage protection measures should follow market economic rules and explore the root causes of conflict from the perspectives of protection motives, property rights relations, and legal institution. The results showed that:

First, in the WCH protection process, the role of the Chinese government is ambiguous, leading to constant conflict between heritage protection and communities. The party secretary of Hangzhou has proposed "better access to West Lake for Chinese and foreign tourists". From an objective point of view, the government protects cultural heritage in order to attract tourists or achieve urban economic growth. In China, national laws lack a clear definition of the powers and responsibilities of central and local governments for protection actions. The government should assume the legal responsibility to protect but needs to define protection for whom. What are the boundaries of the exercise of public power? If the government changes from a "guardian" of cultural heritage to an "operator", then the public power to protect cultural heritage will become a tool for government officials to seek personal gains (urban economic growth is the promotion standard for officials). The conflict of interest in protection actions is actually the lack of definition of the role of the government.

Second, Chinese institution only focuses on the protection of the material structure of cultural heritage and ignores the rights and interests of the local people. On the one hand, this was demonstrated by the series of community control requirements in the WLCLH protection law. China's protection institution only focuses on the protection of cultural resources, but ignores the interest imbalance caused by the restriction of the behavior rights of community residents. On the other hand, China's laws on the protection of WCH are not in agreement with the laws concerning individual property rights; this makes it difficult for community residents to fight for their own legal rights and interests during the protection process.

Third, there is a contradiction between legal ownership and economic behavior rights. From the legal perspective, the ownership of WLCLH belongs to the state. The ownership of local communities belongs to the residents, who have the right to live, buy and sell, repair, and transform. However, from an economic perspective, the state determines the behavior rights to protect the WLCLH through powerful laws, resulting in the "public" characteristics of the communities that were originally privately owned. Therefore, a series of behavior rights of the local residents are restricted and the exclusivity of their behavior rights is violated; this process lacks a fair and legal interest balance mechanism. Thus, it can be seen that there is a paradox between the complete ownership at the legal level and the incomplete behavior rights at the economic level, and the contradiction of rights will certainly lead to a conflict of interest.

Institutional change always maintains the dynamic evolution of "system disequilibrium–system perfection–system equilibrium". Institutional disequilibrium is the premise of institutional innovation [49,64]. The dilemma in China's WCH protection results from the inertia of the planned economy in the establishment of the protection Institution. This only attaches importance to the protection responsibility of cultural heritage resources but neglects the behavior rights, which leads to a loss of the interests of the local people. Therefore, it is necessary to improve the protection system from a "people" perspective and to coordinate the relationships among all parties and create social synergy for the protection of cultural heritage.

First, we should face up to the resource attribute of WCH and rationally view the community's pursuit of economic interests. The characteristics of WCH mean that people's spiritual needs are satisfied far more than their material needs. But it does not mean that the protection institution can force local residents to give up their legitimate economic interests (material needs). Maslow's psychological research shows that people's pursuit of benefits follows a sequence from material benefits to spiritual benefits. For community residents, if the WCH does not result in additional economic benefits, the original legitimate economic benefits should at least not be infringed upon [46]. In modern countries with a market economy, the government recognizes and protects citizens' legitimate rights to pursue economic interests. To understand and accept this point rationally is very important for the improvement of China's WCH protection institution.

Second, the establishment of a protection institution should be connected with the legislation of property rights (Real Right Law) and clearly define the boundary between the government and the community. The protection of WCH inevitably involves ownership and behavior rights. The rights and obligations set for the protection of WCH are bound to affect the interests of the residents. However, as mentioned above, China's legislation on the protection of world heritage culture is separate from legislation concerning individual property rights. The development of the WLCLH has proved that if the rights boundary between the government and community cannot be clearly defined while determining the world heritage protection rights, the lack of a boundary of rights will inevitably lead to a conflict of interest between the government and community residents. To sum up, it is recommended that China uses national legislation to create a new usufructuary right—cultural heritage protection right. The establishment, acquisition, and exercise of this right are clearly stipulated in the legislation to link up the legislation in the field of WCH with the legislation in the field of economy.

Third, administrative means should be used to intervene in the conflicts of interest and form a protection force. Due to the non-exclusive nature of the ownership of WCH resources, protection

has never been an individual task. Different interests and behavior motivation will lead to different actions. In the process of sharing WCH resources, there will be competing interests, and even conflicts of interest. As the manager of WCH, the Chinese government should not only focus on the material structure of WCH, but also observe the stakeholders behind WCH, identify the core interests they care about, predict the consequences of their actions, and then guide more inclusive decision-making processes, improve public or local community consultation, and use better conflict management tactics. Only by balancing the interests among interest subjects can we avoid conflict and promote cooperation.

**Author Contributions:** This manuscript was written by J.L. and X.L. P.Z. contributed to the survey questionnaire and data analysis. All authors read and approved the final manuscript.

**Funding:** The research described in this paper was financially supported by the National Key Research and Development Program of China (Grant Nos. 2018YFD1100805).

**Acknowledgments:** The research described in this paper was financially supported by the National Key Research and Development Program of China (Grant Nos. 2018YFD1100805).

**Conflicts of Interest:** The authors declare that they have no competing interests.

**Data Availability:** Most of the data on which the conclusions of the manuscript rely are published in this paper and the full questionnaire data are available for consultation upon request.

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
