# Peer review of "Rights–Values–Interests: The Conflict between World Cultural Heritage and Community: A Case Study of the West Lake Cultural Landscape Heritage in China"

_sustainability, doi:10.3390/su11174560_

Round 1
Reviewer 1 Report
This paper focuses on an interesting case study and employs a potentially novel approach to issues that have long been discussed in the existing heritage management literature. The following are some suggestions for improvement.
Overall content:
The discussion on what constitutes heritage values (section 2.2.1) would have benefitted from a discussion of the mechanisms behind the heritage valorisation processes and the impact that certain agendas have through the Authorised Heritage Discourse, a term coined by Laurajane Smith (e.g. in her book ‘Uses of Heritage’) but adopted extensively in heritage literature. At the core of the problems in the approaches described in the case study seems to lie the top-down management and top-down hegemony.
The "rights-values-interests" analysis framework (inspired, as mentioned by the authors by the property rights theory of the new institutional economics) seems an interesting take on issues that are relevant to values-based and participatory heritage management, sustainable heritage development etc. I would, however, like to see some stronger arguments articulated on what this framework could add to already existing models discussed in the heritage literature (e.g. the Burra Charter process to involve stakeholders in heritage planning). I would argue that for effectively avoiding the conflicts between state organisations and local communities more than ‘fair and legal "trading mechanisms"’ are required (‘interest balance mechanisms’ seems a more effective term). It seems that cases like WLSA require more inclusive decision-making processes, public or local community consultation, conflict management tactics and other elements prior to imposing laws, regulations, interventions etc.
Conflicts seem to be created by restrictions to house repairs, supply of water and electricity and by the forced relocation of people living in the WCH area. This is clearly presented by the authors. The question however, remains what exactly would heritage protection and tourism development entail for the different stakeholders of the WLSA? In addition, what kind of opportunities exist in terms of tangible, intangible and natural heritage?
Suggestions for clarity of presentation/expression:
The three statements/excerpts on pages 1-2 should be presented as quotes! The same goes for all comments derived from interviews (for example, end of page 8, page 10).
The research questions addressed by the paper should be more clearly clarified in the ‘Introduction’ (section 1) and tied more closely to the research methods (section 3.2).
I’m not very sure if the term ‘heritage community’ as juxtaposed to a ‘normal community’ (p. 5) is explicit enough and a standard/established term!
Improvements in bibliographic support:
I would suggest some citations to relevant bibliography for the statement ‘There is a consensus that property rights theory is important in the field of cultural heritage protection’ (p. 5).
Author Response
Response to Reviewer 1 Comments
Dear expert:
Thank you very much for your valuable suggestions on the paper. I have refined the paper in accordance with your suggestion. The details are as follows:
Point 1: The discussion on what constitutes heritage values (section 2.2.1) would have benefitted from a discussion of the mechanisms behind the heritage valorisation processes and the impact that certain agendas have through the Authorised Heritage Discourse, a term coined by Laurajane Smith (e.g. in her book ‘Uses of Heritage’) but adopted extensively in heritage literature. At the core of the problems in the approaches described in the case study seems to lie the top-down management and top-down hegemony.
Response 1:
The Western philosophy of preservation has been accused of ‘Eurocentricity’ (Jokilehto, 2006), heritage preservation continues to represent the conservative view of the European elite. Smith (2006) labels this conservative and Western discipline the Authorised Heritage Discourse (AHD), which ‘takes its cue from the grand narratives of nation and elite on the one hand, and technical expertise and aesthetic judgment on the other’. Smith further indicate that the AHD is operated from a position of power in order to sustain the privileged positions of a range of experts while simultaneously thwarting or marginalising the interests of others.
Cultural heritage conservation in China has been part of the field of urban planning and has thus been dominated by architects and urban planners (Whitehand and Gu, 2007). The participation of specialities from the other fields has been limited. The discourses of cultural heritage conservation at the central level in China have been prescribed by professional elites and state officials. Therefore, in the process of Authorised Heritage Discourse in China. The government continues to emphasise the importance and significance of the preservation of material culture through the top-down management and top-down hegemony, but exerts less effort to maintain the social contexts of traditional living settlements.
There are some misnomers in the discussion on what constitutes heritage values (section 2.2.1). I have re-written the key content, added references and highlighted core ideas.
Point 2: The "rights-values-interests" analysis framework (inspired, as mentioned by the authors by the property rights theory of the new institutional economics) seems an interesting take on issues that are relevant to values-based and participatory heritage management, sustainable heritage development etc. I would, however, like to see some stronger arguments articulated on what this framework could add to already existing models discussed in the heritage literature (e.g. the Burra Charter process to involve stakeholders in heritage planning). I would argue that for effectively avoiding the conflicts between state organisations and local communities more than ‘fair and legal "trading mechanisms"’ are required (‘interest balance mechanisms’ seems a more effective term). It seems that cases like WLSA require more inclusive decision-making processes, public or local community consultation, conflict management tactics and other elements prior to imposing laws, regulations, interventions etc.
Response 2:
Q1:It is a question arises as to whether the international approach of involving community in heritage conservation has been further applied on the Chinese side in practice. The basic principle of involving communities is to invite them to contribute in the decision-making process. However, given state centralisation, it is hard to imagine that power sharing with local communities would be initiated from the top. Different from the Burra Charter, which places great emphasis on the cultural significance of heritage sites and community consultation, the China Principles ‘reiterate the dominating status of the national heritage law and allow greater room for state intervention in the decision-making procedure’, and ‘focus more on the bureaucratic framework as well as the operational protocol in the conservation process’ (Qian 2007). From the case of WLCLH, local governments also conduct surveys of communities and consult with residents in the heritage management decision-making process. However, the protection laws and regulations and management of WLCLH often ignore the interests and values of local residents. Therefore, it can be argued that community participation is often difficult to fundamentally solve the contradiction, but requires a top-down mechanism to incorporate the legitimate interests and values of local residents into the formulation of protection laws. The "rights-values-interests" analysis framework lays more emphasis on how the government coordinate and implement the public and original values, public and private interests when determining protection behavior rights (formulating protection laws). The "rights" in the analytical framework refers to the behavioral rights in the Property Rights Theory, including ownership right, development right, survival right and so on. Different from other analytical frameworks of cultural heritage, this framework studies the contradiction between heritage protection and community in the context of market economy, and tries to find out the strategies to solve the contradiction from the perspective of the combination of economic law and protection law, implement the community participation proposal into the institution improvement truly. For example, Chinese residents' economic rights such as ownership and development rights are guaranteed by Real Right Law. However, the protection of cultural heritage is often based only on the protection laws, while ignoring the Real Right Law involving residents' economic interests. So I think the innovation of analysis framework is to integrate cultural heritage protection into the market environment for research, and based on the background of China, put forward combining protection law and economic law, to ensure that the Chinese government implements the interests of the residents truly in the process of enacting protection laws, and defines their respective behavior rights boundaries clearly. This was not mentioned by the previous researchers.
I have revised some parts of the discussion and conclusion.
Q2: I have revised “trading mechanisms” into “interest balance mechanisms”.
Q3: I think this is a good suggestion: "more inclusive decision-making processes, public or local community consultation, conflict management tactics". I have added the contents of your suggestion in the last paragraph of the conclusion.
Point 3: Conflicts seem to be created by restrictions to house repairs, supply of water and electricity and by the forced relocation of people living in the WCH area. This is clearly presented by the authors. The question however, remains what exactly would heritage protection and tourism development entail for the different stakeholders of the WLSA? In addition, what kind of opportunities exist in terms of tangible, intangible and natural heritage?
Response 3:
This is an important issue that I have neglected. Although in the statement of “The dynamic process of conflict between WLCLH and local communities” (Title 4), I also mentioned that the protection and tourism development of WLCLH has also had positive impacts and opportunities for the communities. From the reality, the negative impact is indeed greater than the positive impact. But it does not mean that positive impact can be ignored in the discussion. This is unfair and non-objective. So I added the expression of WLCLH's positive impact on the community in the "The WLSA and the local communities were harmonious and mutually reinforcing" (Title 4.1) and the discussion (Title 5). For example, employment opportunities, income growth, and a better natural environment, etc.
Point 4: The three statements/excerpts on pages 1-2 should be presented as quotes! The same goes for all comments derived from interviews (for example, end of page 8, page 10).
Response 4: All comments derived from interviews have been presented as quotes.
Point 5: The research questions addressed by the paper should be more clearly clarified in the ‘Introduction’ (section 1) and tied more closely to the research methods (section 3.2).
Response 5: Three research questions have been re-written. They are more closely to the research methods and conclusions. At the same time, in order to better correspond with the research questions, part of the research methods was also re-written.
Point 6: I’m not very sure if the term ‘heritage community’ as juxtaposed to a ‘normal community’ (p. 5) is explicit enough and a standard/established term!
Response 6: That's good advice. My presentation is too casual. What I want to say is communities have undergone a change from development area communities to protection areas communities. Development area communities means communities that have no restrictions and can develop freely. Protection areas communities means communities are located in areas of heritage and are under restrictions by relevant laws and regulations.
Point 7: I would suggest some citations to relevant bibliography for the statement ‘There is a consensus that property rights theory is important in the field of cultural heritage protection’ (p. 5).
Response 7: I have added three references, all of which are in English.
Most importantly, the author asked an English-speaking expert to improve the language of the full text.
Reviewer 2 Report
The paper deals with an interesting topic about the conflict between world cultural heritage and local community. It is clear and well structured.
- First paragraph (introduction): it is suggested to locate the site geographically (for example by writing in brackets “West Lake (Hangzhou, China)”) in order to immediately introduce the site to the reader
- Line 108: Do you mean ICCROM? ICOMOS?
- Figure 1: it is suggested to put the reference to the figure 1 before the image to avoid having to “go back” during reading.
- There are little typing errors (double spaces or none). For example: Lines 59, 88, 101, 237, 318, 360
- it is suggested to leave at least one space before and after the figures and images to make the text clearer
- The captions of figures 4 and 5 is different from the others
Author Response
Response to Reviewer 2 Comments
Dear expert:
Thank you very much for your valuable suggestions on the paper. I have refined the paper in accordance with your suggestion. The details are as follows:
Point 1: First paragraph (introduction): it is suggested to locate the site geographically (for example by writing in brackets “West Lake (Hangzhou, China)”) in order to immediately introduce the site to the reader.
Response 1: I have added (Hangzhou, China) after West Lake according to your good suggestion.
Point 2: Line 108: Do you mean ICCROM? ICOMOS?
Response 2: Yes, it is ICCROM (International Centre for the Study of the Preservation and Restoration of Cultural Property, Rome). When I typed ICCROM, I accidentally lost the letter "R".
Point 3: Figure 1: it is suggested to put the reference to the figure 1 before the image to avoid having to “go back” during reading.
Response 3: I have put the reference in front of figure 1 to keep readers from going back.
Point 4: There are little typing errors (double spaces or none). For example: Lines 59, 88, 101, 237, 318, 360.
Response 4: I have checked the whole article word for word and improved the “double spaces or none” problem.
Point 5: it is suggested to leave at least one space before and after the figures and images to make the text clearer.
Response 5: I have left at least one space before and after the figures and images in the text.
Point 6: The captions of figures 4 and 5 is different from the others.
Response 6: I have modified the captions format of figures 4 and 5.
Most importantly, the author asked an English-speaking expert to improve the language of the full text.

Reviewer 3 Report
This is a very interesting paper but the English needs serious proof-reading as there are too many sentences or part of sentences that don't make sense
For example,
line39: 'after the accession'
line 42: the full line is to be re-written
line 44: 'prohibit us' who is 'us'
Also, there is a problem with the lack of neutrality that one would expect from a scientific article. Too many 'we believe', 'us', 'prohibit us' hint that the authors are biased.
At last, it would be interesting to have a small section on contested heritage. Maybe see Liu Yang' s last article about it: https://www.tandfonline.com/doi/full/10.1080/02665433.2019.1634638
Author Response
Response to Reviewer 3 Comments
Dear expert:
Thank you very much for your valuable suggestions on the paper. I have refined the paper in accordance with your suggestion. The details are as follows:
Point 1: line39: 'after the accession'
Response 1: I have changed it to "After becoming a world heritage site".
Point 2: line 42: the full line is to be re-written
Response 1: I've re-written the first two lines
Point 3: line 44: 'prohibit us' who is 'us'
Response 3: This is from interviews with local residents, so 'us' refers to local residents. I have already explained who "us" is at the beginning of this paragraph, and indicated the interviewee and time in the brackets at the end of this paragraph.
Point 4: Also, there is a problem with the lack of neutrality that one would expect from a scientific article. Too many 'we believe', 'us', 'prohibit us' hint that the authors are biased.
Response 4: I have checked the whole article word for word and re-written all statements with 'we believe', 'us', 'prohibit us'.
Point 5: At last, it would be interesting to have a small section on contested heritage. Maybe see Liu Yang' s last article about it:
https://www.tandfonline.com/doi/full/ 10.1080/02665433.2019.1634638
Response 5: I think this is a very enlightening article, which can be a good supplement to my existing theoretical review. So I included it in my references. In addition, it provides a good idea for another article of mine.
Most importantly, the author asked an English-speaking expert to improve the language of the full text.
